# Hydrotalcite-Embedded Magnetite Nanoparticles for Hyperthermia-Triggered Chemotherapy

**DOI:** 10.3390/nano11071796

**Published:** 2021-07-09

**Authors:** Konstantinos Simeonidis, Efthimia Kaprara, Pilar Rivera-Gil, Ruixue Xu, Francisco J. Teran, Evgenios Kokkinos, Athanassios Mitropoulos, Nikolaos Maniotis, Lluis Balcells

**Affiliations:** 1Department of Chemical Engineering, Aristotle University of Thessaloniki, 54124 Thessaloniki, Greece; kaprara@auth.gr; 2Ecoresources P.C., Giannitson-Santaroza Str. 15-17, 54627 Thessaloniki, Greece; kokkinos@ecoresources.gr; 3Integrative Biomedical Materials and Nanomedicine Lab, Universitat Pompeu Fabra, 08003 Barcelona, Spain; pilar.rivera@upf.edu (P.R.-G.); ruixue.xu01@estudiant.upf.edu (R.X.); 4IMDEA-Nanociencia, Ciudad Universitaria de Cantoblanco, 28049 Madrid, Spain; francisco.teran@imdea.org; 5Nanobiotecnología (iMdea-Nanociencia), Unidad Asociada al Centro Nacional de Biotecnología (CSIC), 28049 Madrid, Spain; 6Hephaestus Advanced Laboratory, Department of Chemistry, International Hellenic University, 65404 Kavala, Greece; amitrop@chem.ihu.gr; 7Department of Physics, Aristotle University of Thessaloniki, 54124 Thessaloniki, Greece; nimaniot@physics.auth.gr; 8Institut de Ciencia de Materials de Barcelona, CSIC, 08193 Bellaterra, Spain; balcells@icmab.es

**Keywords:** layered double hydroxide, Fe_3_O_4_, continuous flow synthesis, magnetic hyperthermia, nanocomposite, drug delivery, cell internalization

## Abstract

A magnetic nanocomposite, consisting of Fe_3_O_4_ nanoparticles embedded into a Mg/Al layered double hydroxide (LDH) matrix, was developed for cancer multimodal therapy, based on the combination of local magnetic hyperthermia and thermally induced drug delivery. The synthesis procedure involves the sequential hydrolysis of iron salts (Fe^2+^, Fe^3+^) and Mg^2+^/Al^3+^ nitrates in a carbonate-rich mild alkaline environment followed by the loading of 5-fluorouracil, an anionic anticancer drug, in the interlayer LDH space. Magnetite nanoparticles with a diameter around 30 nm, dispersed in water, constitute the hyperthermia-active phase able to generate a specific loss of power of around 500 W/g-Fe in an alternating current (AC) magnetic field of 24 kA/m and 300 kHz as determined by AC magnetometry and calorimetric measurements. Heat transfer was found to trigger a very rapid release of drug which reached 80% of the loaded mass within 10 min exposure to the applied field. The potential of the Fe_3_O_4_/LDH nanocomposites as cancer treatment agents with minimum side-effects, owing to the exclusive presence of inorganic phases, was validated by cell internalization and toxicity assays.

## 1. Introduction

Magnetic fluid hyperthermia (MH) has been developed as an alternative approach for the heat-mediated treatment of cancer cells at a controllably localized level [1]. MH stands on the energy losses of magnetic nanoparticle (MNP) dispersion subjected to AC magnetic fields (H_AC_), resulting in a temperature elevation of the dispersion medium. In recent years, significant research effort has been devoted to the synthesis of MNPs with high heating efficiency, their successful incorporation into biological matrices (cells or tissues), the treatment optimization based on theoretical models and the technical improvement of field generation devices and error-free measuring protocols [2,3]. Thanks to their facile and low-cost availability in various geometries, their chemical stability, affordable biocompatibility and magnetic response, iron oxide nanoparticles are widely considered as the most efficient agents for magnetic hyperthermia applications [4,5]. The best heating efficiency for such systems is usually achieved for particle dimensions in the range of the magnetic monodomain region [6], where hysteresis losses are maximized, whereas the positive role of shape adjustment and long-scale arrangement has been demonstrated [7,8]. However, a major drawback for the wide adoption of magnetic hyperthermia in clinical use is the failure to validate the high heating rates of preliminary tests when nanoparticles are placed into biological environments [9]. Such inconsistency is generally attributed to the intracellular immobilisation of particles, aggregation effects on magnetic losses, heat flow dynamics or chemical modification of nanoparticles and degradation effects caused by the dissolution activity of buffer solutions on solid phases [10].

Still, systemic therapies such as those using one or more anticancer drugs introduced into the blood stream, remain in the front line of treatment strategies. Chemotherapy is normally based on the cytotoxic effect of specific molecules which inhibit the division of rapidly growing cells, such as the cancer ones. Unfortunately, the activity of these drugs is not confined to the cancer cells but they can harm any other healthy tissue type with a high dividing rate. Apart from relevant side-effects, another disadvantage of curative chemotherapy is its inability to localize its therapeutic action only on the tumor area, which implies the need of increasing the effective dose of drugs, while, as mentioned above, it can also harm healthy tissues. In order to improve the efficiency of chemotherapy, its conjunction with other modalities such as surgery, radiation and hyperthermia therapy is commonly applied.

Recently, the combination of magnetic hyperthermia with simultaneous drug delivery has been widely studied as an adjuvant therapy to enhance the permeability of anticancer molecules flowing into the blood [11]. This is generally implemented by the attachment of drug molecules onto the surface of magnetic nanoparticles which enables their direction to a specific site by the proper application of an external magnetic field. The grafting of anticancer molecules onto nanoparticles is commonly realized through chemically adsorbed organic compounds including natural biopolymers such as chitosan, polysaccharides, cellulose, lipids, micelles and copolymers. The loading capacity of these ligands with respect to the nanoparticle mass is relatively low and therefore, losses during transfer become significant. Additionally, in the effort to achieve the required dose, sometimes toxicity issues may arise.

Inorganic phases (silica, carbon, metal hydroxides) have also been proposed as drug loading agents indicating higher loading capacities, good stability and biocompatibility [12,13]. Their ability to host anticancer drugs lies on the high surface charge which facilitates strong binding into the crystal structure. More specifically, layered double metal hydroxides (LDH) with multilayer structure provide an ideal substrate to stabilize large quantities of anionic forms taking advantage of their whole volume and not just the surface area [14,15,16,17]. Beyond the significant increase of loading capacity for pharmaceutic molecules, LDH are preferable also for their low synthesis cost through aqueous precipitation methods from metals salts. The general formula of LDH is [Μ^2+^_1−x_Ν^3+^_x_(OH)_2_]^x+^(A^m−^)_x/m_⋅nH_2_O, M: bivalent metal, Ν: trivalent metal, A: interlayer anions, m and n integer numbers, and x a number between 0–1 [18]. Several studies report the substitution of the interlayer anions by anti-inflammatory, cardiovascular and anticancer drugs either by introducing the pharmaceutics during synthesis or by postpreparation ion-exchange and reconstruction [16,19,20]. The most studied cases are Mg/Al and Zn/Al LDH stabilized with carbonate, chloride or nitrate ions [21,22,23].

Hybrid magnetic/LDH nanostructures loaded with therapeutic drugs have been developed with the aim of promoting targeted drug delivery [24]. For instance, MgFe_2_O_4_ nanoparticles were coated by a Mg-Al-NO_3_ LDH and then loaded with ibuprofen or glucuronate [25]. Magnesium ferrite nanoparticles combined with Mg-Al or Zn-Al LDH were also tested as host systems for 5-aminosalicylic acid, diclofenac and ibuprofen [26,27,28]. In several other examples, Fe_3_O_4_ nanoparticles served as the Mg-Al LDH substrate and the system was loaded with fluvastatin and ibuprofen [29,30]. For the same nanocomposite, the loading capacity for anticancer doxifluridine was significantly high, reaching 9.7% [31]. In all these cases, magnetic featuring of the drug-loaded nanocomposites was only used for the magnetically assisted delivery through the application of a static magnetic field.

Here, we report an attempt to illustrate the potential incorporation of inorganic magnetic nanohybrids, consisting of Fe_3_O_4_ nanoparticles and Mg-Al LDH loaded with anticancer 5-fluorouracil (C_4_H_3_FN_2_O_2_, FU), as a way to improve therapeutic efficiency by combining magnetic hyperthermia and chemotherapy. Importantly, a major milestone was to go beyond the parallel occurrence of the two therapeutic modalities as described elsewhere for doxorubicin-loaded nanocomposites [32], and provide their coupling, i.e., the drug release switch-on upon application of the AC magnetic field (Figure 1). The main advantage of the proposed nanocomposites is their fully inorganic nature which is able to combine the heating capability of magnetic nanoparticles with the drug hosting capacity of the layered double hydroxide, considering also that the Mg/Al layered double hydroxide is already recognized for its compatibility for human use as an antacid.

## 2. Materials and Methods

### 2.1. Nanocomposite Synthesis

The synthesis of the nanocomposite consisting of Fe_3_O_4_ nanoparticles (IONPs) distributed into a matrix of Mg/Al layered double hydroxide was carried out in a continuous-flow sequence of two stirring reactors (operating volume 1 L) by the combined precipitation of iron, magnesium and aluminium salts (Appendix A). In the first reactor, Fe_3_O_4_ seeds were prepared after the coprecipitation of FeSO_4_·7H_2_O and Fe_2_(SO_4_)_3_·9H_2_O, which were pumped in the form of aqueous solutions (5 mM), under alkaline conditions (pH 11) regulated by the continuous addition of NaOH solution (2 g/L) in drops. The black-coloured suspension was directed into the second reactor in which the coprecipitation of Mg(NO_3_)_2_·6H_2_O and Al(NO_3_)_3_·9H_2_O took place. The two reagents were pumped as aqueous solutions with concentration 10 mM and hydrolyzed at a pH 9 maintained by the addition of a 1:1 mixture of NaOH/Na_2_CO_3_ (3.5 g/L). Sodium carbonate was introduced in this step to serve also as the source of CO_3_^2−^ which participated in the building up of the layered hydrotalcite structure. Each reactor operated with a residence time of 1 h. The final product was received in the outflow of the second reactor and then, centrifuged and washed several times to remove soluble residuals. The described reactions can be realized with similar success in batch reactors, however, advantages such as the good reproducibility, the achievement of constant concentrations in all ionic and solid forms, the minimization of operation cost and the scale-up potential to mass production would not be covered. A schematic summary of the process is presented in Figure 2 while a picture of the laboratory continuous-flow system appears in Appendix A.

The weight percentage of magnetite nanoparticles in the nanocomposite and the Mg-to-Al molecular ratio in the layered double hydroxide were determined by properly modifying the reagents’ flowrates. For instance, to receive a nanocomposite with 35 wt.% in Fe_3_O_4_ nanoparticles (MagnoTher, MGT-35), both FeSO_4_ and Fe_2_(SO_4_)_3_ flowrates were adjusted to 0.1 L/h while Mg(NO_3_)_2_ and Al(NO_3_)_3_ were added with a flowrate of 0.24 and 0.08 L/h, respectively. To decrease Fe_3_O_4_ content at 20 wt.% (MagnoTher, MGT-20), Mg(NO_3_)_2_ and Al(NO_3_)_3_ flowrates were increased to 0.45 and 0.15 L/h, respectively. Using such proportions, the Mg-to-Al molecular ratio was adjusted to 3. Under such conditions, the production rate of the nanocomposite in terms of dry solid varied between 0.35−0.55 g/h.

### 2.2. Characterization

An overview of produced nanocomposites’ morphology and separately of their constituting phases was obtained by electron microscopy. For more clarity on the distribution of IONPs, high magnification images were taken by transmission electron microscopy (TEM) using JEM-1210 (JEOL, Tokyo, Japan), operating at 120 kV. TEM samples were prepared by dropping a diluted aqueous dispersion of the material onto a carbon coated copper grid. Quasi-static magnetic properties of the samples were measured using a superconducting quantum interference device (SQUID) MPMS XL-7T magnetometer (Quantum Design, San Diego, CA, USA). Structural-phase identification was performed by powder X-ray diffractometry (XRD) using a water-cooled Ultima+ diffractometer (Rigaku, Tokyo, Japan) with CuKa radiation, a step size of 0.05° and a step time of 3 s, operating at 40 kV and 30 mA.

Average elemental content of the nanocomposites was determined by graphite furnace atomic absorption spectrophotometry, using a AAnalyst 800 instrument (Perkin Elmer, Waltham, MA, USA). The actual ratio of Fe^2+^/Fe^3+^ in the Fe_3_O_4_ fraction was defined after digestion of a weighted quantity of each sample in 7 M H_2_SO_4_ under heating and titration with 0.05 M KMnO_4_ solution till the appearance of a pink colour. The percentage of carbonates (CO_3_^2−^) located in the interlayer space of hydrotalcite was quantified using a FOGL bench-top soil calcimeter (BD Inventions, Thessaloniki, Greece) with a determination error of less than 5%.

The potentiometric mass titration method was applied to define the positive charge density of the solid. In the first step, the point of zero charge (PZC) was determined after equilibrating water suspensions of the nanocomposites (10 g/L) in 0.001, 0.01 and 0.1 M NaNO_3_ solutions and adjusting pH to 11 by adding 0.1 M NaOH. Then, suspensions were titrated by adding stepwise small quantities of a 0.1 N HNO_3_ solution and recording equilibrium pH until pH 3 was reached. For the three ionic strengths, the plotting of surface charge density, which is proportional to the difference of acid volume used to set the same pH in the dispersion and a blank titration, indicates the PZC as the point of intersection.

### 2.3. Magnetic Heat Losses

Calorimetry measurements of magnetic suspensions under H_AC_ were performed using a commercial AC magnetic field generator (SPG–06-III 6 kW High Frequency Induction Heating Machine, Shenzhen Shuangping Ltd., Shenzhen, China) working at 765 kHz frequency and 24 kA/m magnetic field intensity. The specific loss power (SLP), referred to as specific absorption rate (SAR) hereafter, was derived from the slope of the temperature versus time curve after subtracting water background signal and heat losses to the environment [33].

Temperature variations during the application of AC field were recorded using a commercial optical fibre thermal probe located in the centre of the sample and connected to an PicoM device (Opsens, Quebec, QC, Canada) with an experimental error of ±0.1 °C. SAR values under non-adiabatic conditions were determined through the temperature increment as a function of time (dT/dt) using the following expression:(1)$$\mathrm{SAR}=\frac{C_{d}m_{d}}{m_{\mathrm{Fe}}}\frac{\mathrm{dT}}{\mathrm{dt}}$$
where C_d_ is the mass specific heat of the dispersion media, m_d_ is the dispersion’s mass, m_Fe_ is the iron mass related to the IONPs diluted in the dispersion and dT/dt is the effective slope upon switching H_AC_ on, after subtracting the contributions of coil surface heating and environment cooling. The value of C_d_ considered in this study was 4.18 J/g K for water dispersion.

AC magnetometry measurements of the magnetic colloids were carried out by commercial inductive magnetometers (AC Hyster Series; Nanotech Solutions, Madrid, Spain). The AC Hyster Series magnetometer offers a wider field frequency range from 10 kHz up to 300 kHz and field intensities up to 24 kA/m which are automatically selected. Hyster Series measure magnetization cycles from IONPs dispersed in liquid media at room temperature, consisting of three repetitions to obtain an average of the magnetization cycles and the related magnetic parameters (H_C_, M_R_, Area). In order to accurately quantify the magnetic losses of IONP suspensions, the specific absorption rate (SAR) values were calculated according to SAR = *A*⋅*f*, where *A* is the magnetic area and *f* is AC magnetic field frequency [34].

### 2.4. Drug Loading and Release

Loading of 5-fluorouracil in the interlayer space of the Mg-Al LDH part of the nanocomposite was carried out by equilibrating a quantity of the samples with a solution of the drug in phosphate-buffered saline (PBS). The obtained FU-loaded nanocomposite sample is referred as Fe_3_O_4_/LDH-FU. In these experiments, a freshly prepared 20 mM stock solution of 5-fluorouracil in PBS was used after proper dilution. Exchange between 5-fluorouracil molecules and carbonates was maximized when experiment took place at pH 9. In these conditions, the kinetic behavior of loading was studied using a 2 g/L dispersion of MGT-35 in a 0.5 mM 5-fluorouracil PBS solution and measuring the residual concentration in different time intervals till equilibrium was reached. The adsorption capacity variation (isotherm) as a function of residual 5-fluorouracil was determined after equilibrating 2–20 g/L of nanocomposite with a 15 mM drug solution.

After drug loading, the leaching behavior under different pH values was evaluated. Sample loaded with around 1 mmol/g of 5-florouracil was dispersed in PBS adjusted to pH 4.0, 7.4 and 8.5, and the released drug was monitored for a period of up to 60 min. Similarly, the release of drugs was examined for various temperatures, 10, 20 and 35 °C, while in another experiment the temperature of the sample was increased at 40 °C by the application of AC magnetic field and kept at this value for 10 min.

### 2.5. Cell Internalization

Human colon cancer cells (HT29) were cultured in a microscope coverslip placed in the wells of a 12-well plate containing DMEM/F-12 (Dulbecco’s Modified Eagle Medium/Nutrient Mixture) basal medium supplemented with 10% fatal bovine serum (FBS), 1% l-glutamine and 1% penicillin/streptomycin. After 24 h, they were incubated with the nanoparticles (IONPs, Fe_3_O_4_/LDH and Fe_3_O_4_/LDH-FU) at a concentration of 0.1 mg/mL in growth media for two days. Then, the cells were washed and incubated with lysotracker red (0.25 µM dispersed in basal media for 25 min) to stain the lysosomes. The cells were then washed with PBS, dispersed in PBS and observed under a LSP2 Leica Confocal Laser Scanning Microscope. Samples were excited with a 543 nm Green Helium-Neon laser and collected emitted light from 555 nm to 620 nm.

### 2.6. Toxicity

A resazurin-based cytotoxicity assay was used for checking the biocompatibility of the nanoparticles (IONPs and Mg-Al LDH-FU) in HT29. A number of 20,000 cells/well in 100 µL growth medium were seeded in a 96-well plate. After growth for two days, cells were incubated with different nanoparticle concentrations ranging from 0 to 256 mg/mL for one day. As negative control for cell viability, cells were incubated with dimethyl sulfoxide (DMSO) inducing cell death. Three wells were used per sample tested. Afterwards, resazurin (10%) was added for three hours and measured for its fluorescent metabolite (resorufin) (λ_ex_ 560 nm; λ_em_ 572–650 nm). First, the maximum intensity emission wavelength value was taken and the maximum intensity value obtained for the negative control to remove background signal was subtracted. Then, data was normalized with the highest value. Finally, the normalized values (relative cell viability, %) were plotted towards the logarithm of the concentrations. GraphPad Prism 6 was used to obtain the logIC_50_. This experiment was replicated three times.

## 3. Results and Discussion

### 3.1. Material Properties

The structural characterization of the developed nanocomposite indicated the preservation of the constituting phases, LDH and iron oxide, in the final product. Figure 3 shows the XRD diagrams of the pure components when separately prepared in comparison to their sequential preparation as a hybrid in the two-stage reaction setup. The observed diffraction peaks for IONPs fitted well to those expected for the inverse spinel structure of Fe_3_O_4_ whereas the layered formation of the Mg-Al LDH was signified by the strong low-angle peaks which were identified as hydrotalcite. Chemical analysis indicated that Mg/Al molecular ratio was 2.9/1 which is very close to the nominal value for the R3m space group of hydrotalcite. For initially formed hydrotalcite, the unit cell parameters of its 3R stacking sequence were calculated to be α = 0.3055 nm and c = 2.2725 nm respectively, whereas the interlamellar distance estimated by the reflection (003) was found to be 0.7755 nm. Such a value is determined by the hosted structural carbonates which reached a percentage of 14.0 wt.% in the Mg-Al LDH.

The presence of Fe_3_O_4_ instead of γ-Fe_2_O_3_ as the dominant iron oxide was verified after determination of the Fe^2+^/Fe^3+^ ratio at around 0.41 very close to the expected stoichiometry for magnetite (0.5). Overall, the chemical analysis indicated the percentages of Fe: 22.7 wt.%, Mg: 13.5 wt.%, Al: 7.2 wt.% for sample MGT-20 and Fe: 39.8 wt.%, Mg: 11.0 wt.%, Al: 6.1 wt.% for sample MGT-35 which came in good agreement to the targeted mass ratio for the LDH and Fe_3_O_4_ phases in the nanocomposites.

The nanocomposites showed relatively high specific surface areas, 78 m^2^/g for MGT-20 and 65 m^2^/g for MGT-35, although they appeared significantly decreased in comparison to the pure Mg-Al LDH (175 m^2^/g). The nanocomposites indicated also a significant surface charge density which was maintained at around 0.7 mmol OH^−^/g.

The TEM images shown in Figure 4 provide a representative view on the nanoscale morphology and distribution of the nanocomposites in comparison to the separately prepared pure LDH and IONPs. Following the described precipitation method, Mg-Al LDH form very thin nanosheets, while IONPs have a nearly spherical shape with an average diameter of 31 ± 6 nm. The sequential synthesis of the two phases resulted in a good distribution of the nanoparticles onto the surface of the LDH sheets. Magnetic interactions between nanoparticles and the absence of any stabilizing agent during synthesis contributed to the observed aggregation effects. The lower magnification images of the sample MGT-35 (Appendix A), recorded with scanning electron microscopy, indicate that in the powder form, the layered matrix appears as a continuous substrate without obvious limits of the separation that occurs when dispersed. This effect may be attributed to the secondary self-organization of the layered structure by weak forces during dewatering and it is fully reversible considering that a hydrodynamic diameter of samples was measured around 500 nm.

The magnetic response of the nanocomposites was attributed to the participation of Fe_3_O_4_ and its intensity appeared to be proportional to the magnetic phase percentage. The hysteresis loops under quasi-static conditions shown in Figure 5 indicated that samples MGT-20 and MGT-35 had saturation magnetisation values around 17 and 30 Am^2^/kg respectively, in good accordance to the measured magnetisation for pure IONPs (90 Am^2^/kg) and the mass percentage of Fe_3_O_4_ in each case.

### 3.2. Hyperthermia Performance

The temperature increase during the AC field application indicated a significantly high potential of the samples to deliver heat flow to the environment (Appendix A). For instance, a temperature increase from room temperature to around 35 °C within 2 min of field application was obtained for a 2 g/L aqueous dispersion of sample MGT-35, yielding a SAR value of 1970 W/g_Fe_ (±10%). Keeping the same field strength, SAR values seemed to follow an exponentially increasing trend in the frequency range from 30 to 765 kHz succeeding in relatively high performances even for frequencies below 100 kHz (Figure 6). The uncertainty of the determined SAR values in the lower frequencies range was quite small (typically below 3%) following the accuracy of the AC magnetometry measurements. Importantly, it was found that the addition of the Mg-Al LDH phase did not appear to modify the heating performance of IONPs and therefore, temperature rise was proportional to the content of the magnetic phase.

Compared to other studies on single iron oxide nanoparticles or LDH composites, the obtained SAR values were among the highest reported, covering a very wide frequency range. Typically, Fe_3_O_4_ nanoparticles prepared by the oxidative precipitation method, showed efficiency of around 2 kW/g at 765 kHz, translated into 2.3 W/g_Fe_ when produced by a continuous flow process [35], but less than half when produced in batches [36]. Aiming to achieve higher SAR values (up to 10 kW/g at 500 kHz), combined ferrite phases were employed but the requirement for using hazardous reagents for their synthesis and the toxicity of elements such as Mn and Co inhibit their potential for clinical application [37,38,39]. Research reports on the heating performance of magnetic nanoparticle-decorated LDH systems are only scarce. For example, Fe_3_O_4_/Mg-Al LDH nanohybrids were found to reach an SAR of 73.5 W/g at 425 kHz and 30 kA/m [32]. The same study provided promising results concerning the combined hyperthermia and drug delivery with doxorubicin as well as the therapeutic efficiency in HeLa cells.

### 3.3. Drug Release Behavior

The capacity of the Mg-Al LDH structure to host 5-fluorouracil molecules after exchange with structural carbonates was first demonstrated through the kinetic experiments (Appendix A). Within less than 1 h, MGT-35 was able to capture significant quantities from the equilibrated drug solution overcoming a loading of 6.5 mmol/g and approaching the 75% of its maximum ability (~8.7 mmol/g) into PBS at pH 9. The procedure is described as:Mg_6_Al_2_(OH)_16_(CO_3_^2−^) + C_4_H_3_FN_2_O_2_^−^ → Mg_6_Al_2_(OH)_16_(C_4_H_3_FN_2_O_2_^−^) + CO_3_^2−^

It should be noted that the loss of carbonate content at the end of this experiment, which validates the presence of the ion exchange mechanism, was around 1 wt.% Considering the uptake capacity values for 5-fluorouracil, carbonate losses appeared much higher than the stoichiometrically expected ones, suggesting that the incorporation of drug’s voluminous molecule caused the release of multiple carbonate ions in order to fit in the interlayer space. The partial replacement of the structural carbonates by 5-fluorouracil was also reflected in the expansion of Mg-Al LDH unit cell which was revealed by the shift of the XRD diffractogram to smaller angles in the drug-loaded sample (Appendix A). The uptake capacity can be adjusted by varying the dispersion’s concentration and therefore, the residual 5-fluorouracil concentration as shown in the adsorption isotherm of Appendix A.

The stability of loaded 5-fluorouracil and the release rate can be evaluated by modifying the pH of the dispersion medium (Appendix A). At pH 8.5, slightly below the loading acidity, the release rate was very low with less than 20 wt.% of the drug to be found in soluble state. However, the loss percentage within 1 h of contact, reached 50 wt.% when the pH was adjusted to 7.4. The weakening of the layered structure by the increase of metal component solubility was the reason for the increasingly observed drug release. Under more acidic conditions, the whole quantity of drug was completely released immediately, however, a significant part was captured back to the nanocomposite within less than 1 h of contact. The last observation could be of high importance when stimuli-responsive drug delivery systems are required.

The temperature of the studied medium was another important parameter which defined the release rate even in the short term (Figure 7). More specifically, the loading loss at temperatures of 10, 20 and 35 °C (close to the human body temperature) after 5 h of contact at pH 7.4 was found to be 45, 55 and 70 wt.%, respectively. The temperature-dependence study provides a general view of the release kinetics in a wide temperature range indicating the behavior of the drug-loaded nanocomposite during storage in a refrigerator or its application in cancer treatment. Importantly, even at the higher temperature, around 20% of the initial load was stabilized into the nanocomposite for several hours. A very interesting finding that validates the motivation of this study was that the drug release was very rapid and reached 80 wt.% only by applying the AC magnetic field for 10 min and reaching a temperature of 40 °C by means of magnetic hyperthermia. This result should be attributed to the high localized temperature increase within the nanocomposite mass which was able to initiate a fast release response to the LDH phase.

### 3.4. Cellular Uptake and Biocompatibility

The biological characterization of the pure and drug-loaded MGT-35, in comparison to the corresponding IONPs, involved cell internalization and the induced cellular toxicity. The physicochemical properties of nanoparticles (e.g., size, shape, charge) are known to affect cellular responses such as internalization (rates and mechanisms) or cytotoxicity [40,41,42,43] and, therefore, knowledge of these parameters is crucial to predict the nanoparticle potential for the magnetic hyperthermia treatment of cancer in real biological conditions that may interfere with performance. For example, previous results showed differences in the uptake of magnetic nanoparticles depending on the surface charge (positive vs. negative) [44]. Quantitative analysis indicated that positively charged magnetic nanoparticles avoid the early endosomes and they are preferentially located in the lysosomes. On the other hand, negatively charged nanoparticles were first accumulated in early endosomes and then, transferred to the lysosomes with time. In general, positively charged nanoparticles are considered as more toxic (at least, acutely) than their negatively charged counterparts due to their stronger interaction with cellular components [43,45].

HT29 cells were incubated with the three tested systems, i.e., Fe_3_O_4_ nanoparticles, pure MGT-35 and drug-loaded MGT-35 (MGT-35-FU). Figure 8 shows that the nanoparticles were steadily taken up by the cells located in the lysosomes after two days of incubation. The nanomaterials provided a high contrast in the transmitted channel of the confocal microscope, mostly due to a strong intracellular accumulation. This high contrast enabled their tracking inside the cells and colocalization inside the lysosomes (labelled in red). Figure 9 illustrates the toxicity profile at the level of mitochondrial activity of the three samples when exposed at different concentrations ranging from 256 to 0.5 mg/mL for 24 h. Reference IONPs practically exhibited no toxicity at the given concentrations and exposure time. Iron oxide nanoparticles are known to be metabolized with the iron metabolism in the spleen and liver [46]. MGT-35 and MGT-35-FU showed a sigmoidal curve for toxicity. The IC_50_ values of Fe_3_O_4_, MGT-35 and MGT-35-FU, which indicate the concentrations needed for each system to kill half of the cell population, were 476 mg/mL, 12 mg/mL, and 18 mg/mL, respectively. It appears that the presence of the Mg-Al LDH modifies the toxicity profile of the sample, although the provided concentration range for safe use without chemically induced side-effects still remains wide. Such observations should be attributed to the relatively good structural stability which favours membrane damage and the higher release of metal ions and carbonates at the acidic conditions occurring in endosomes or lysosomes [47]. Noteworthy, MGT-35-FU and MGT-35 exhibited similar IC_50_ values. Their toxic profile appeared to be dominated by the Mg/Al layered double hydroxide matrix while the effect of the 5-fluorouracil presence was minor. Such a finding was attributed to the drug release of around 80% (Figure 7) which took place during the initial washing of the nanocomposite before contact with the cells. The remaining 20% could be stabilized into the structure for the time window of the experiment, as explained by the temperature-dependent kinetic data. The application of localized heating by magnetic hyperthermia enabled this barrier to be overcome and rapidly promoted the complete release of this fraction which appeared to be the most strongly captured. Another possible explanation is that the presence of serum protein corona around the nanoparticles may influence the short-term release kinetics of 5-fluorouracil [41,44].

## 4. Conclusions

A 100% inorganic-based drug carrier was developed to be used for the controllable delivery and release of anticancer molecules. In particular, a nanocomposite built from iron oxide nanoparticles and a matrix of Mg/Al layer double hydroxide was produced by an environmentally friendly and scalable procedure which provides very stable synthesis conditions, high production rates and an affordable cost for an engineered drug-delivery system. The nanocomposite is capable of loading up to several mmol/g of anionic molecules, such as 5-fluorouracil, which is then released very rapidly when an AC magnetic field is externally applied, due to the high heat generation at the microstructure level. Importantly, such advantages are combined with sufficient cell internalization of the nanocomposite and very limited toxicity, even for relatively high applied concentrations.

## Figures and Tables

**Figure 1 nanomaterials-11-01796-f001:**
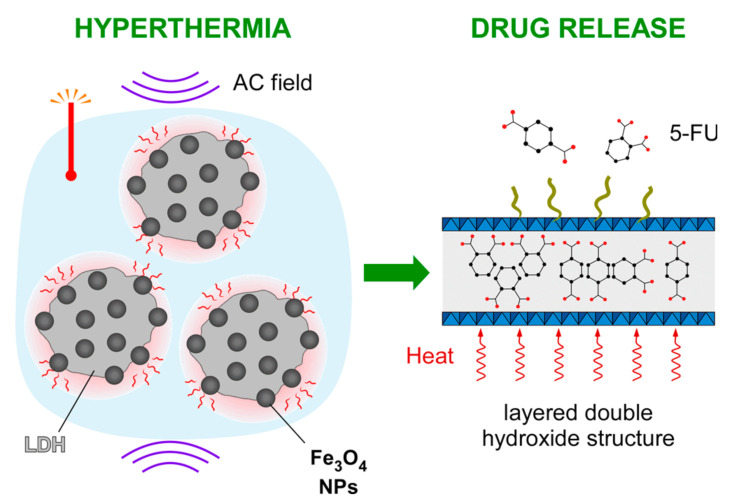
Graphical concept of controllable 5-fluorouracil release from the Fe_3_O_4_/Mg-Al LDH nanocomposite when exposed to a radiofrequency AC magnetic field.

**Figure 2 nanomaterials-11-01796-f002:**
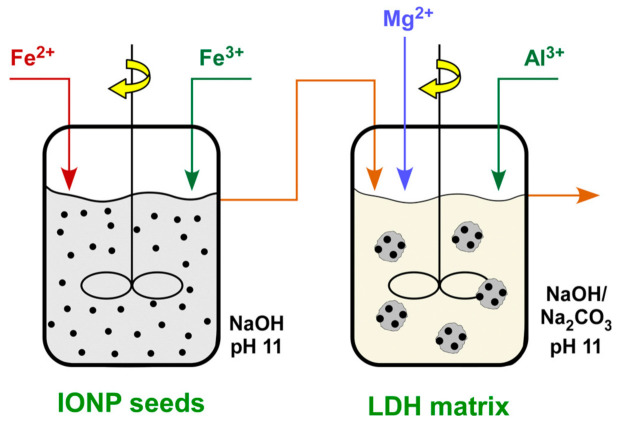
Continuous-flow reactor sequence for the production of the Fe_3_O_4_/Mg-Al LDH nanocomposite.

**Figure 3 nanomaterials-11-01796-f003:**
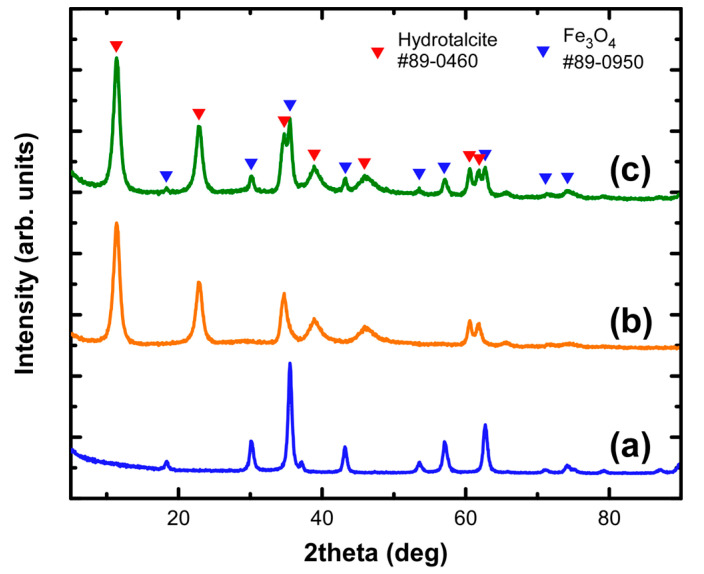
XRD diagrams of the IONPs (**a**), pure hydrotalcite (**b**) and MGT-35 (**c**).

**Figure 4 nanomaterials-11-01796-f004:**
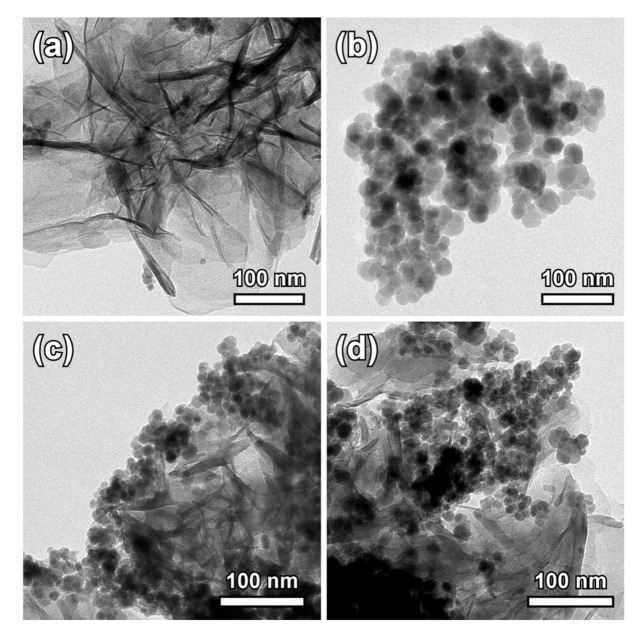
TEM images of the pure hydrotalcite phase (**a**) and the IONPs (**b**). Corresponding images showing the nanostructure of MGT-20 (**c**) and MGT-35 (**d**).

**Figure 5 nanomaterials-11-01796-f005:**
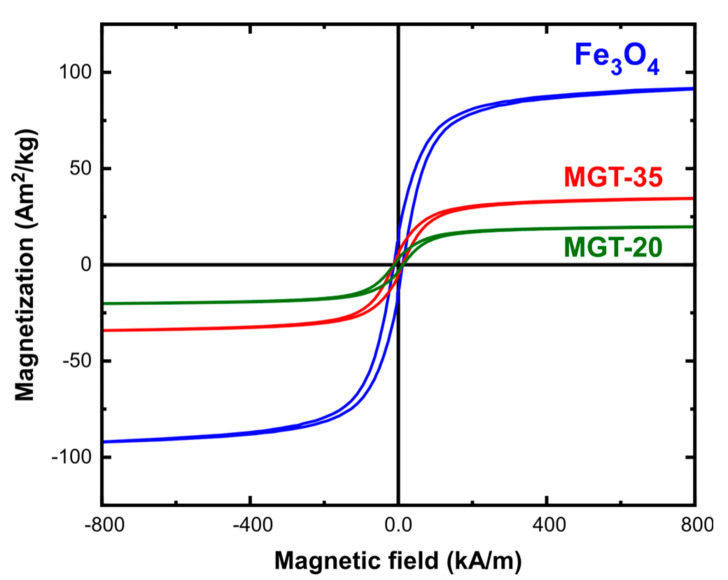
Magnetic hysteresis loops of MGT-20 and MGT-35 in comparison to the constituting IONPs.

**Figure 6 nanomaterials-11-01796-f006:**
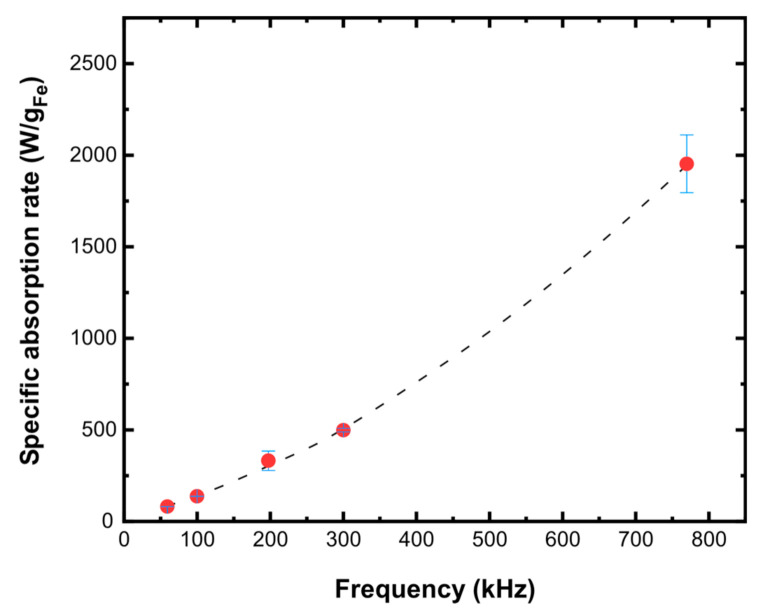
SAR values determined at various AC magnetic field frequencies and strength 24 kA/m for aqueous dispersions (2 g/L) of MGT-35. Dashed line is a guide to the eye.

**Figure 7 nanomaterials-11-01796-f007:**
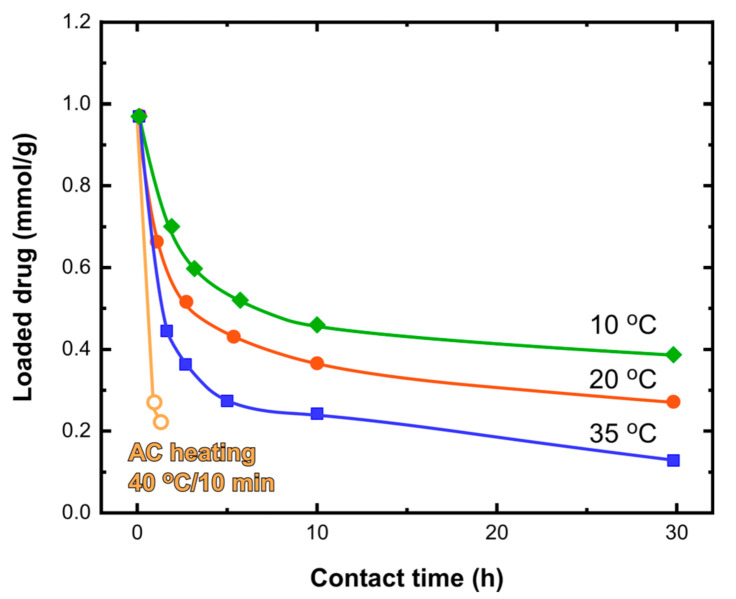
Drug release evolution for drug-loaded MGT-35 in PBS solution adjusted to pH 7.4 for various temperatures and corresponding curves for keeping the dispersion at 40 °C by applying AC magnetic field.

**Figure 8 nanomaterials-11-01796-f008:**
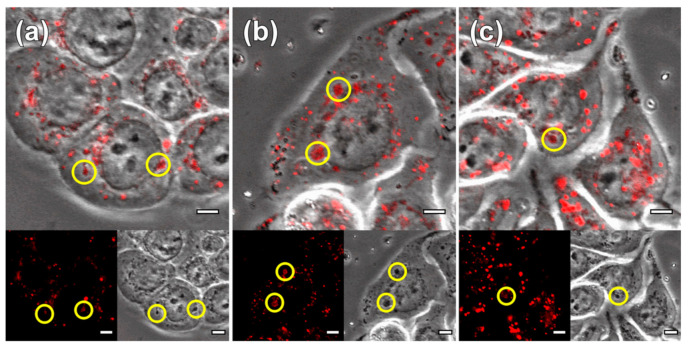
Internalization of Fe_3_O_4_ nanoparticles, (**a**) pure MGT-35 (**b**) and MGT-35-FU (**c**) by HT29 cells. Cells were incubated with 0.1 mg/mL nanoparticles for two days. The nanoparticles are visible due to the high contrast provided in the transmission channel (low right inset) The lysosomes of the cells were labelled with lysotracker (red dots) (low left inset). The main image corresponds to the overlay of the fluorescent (red) and transmission channel. The yellow circles mark the overlap of nanoparticles and lysosome. The scale bar represents 5 µm.

**Figure 9 nanomaterials-11-01796-f009:**
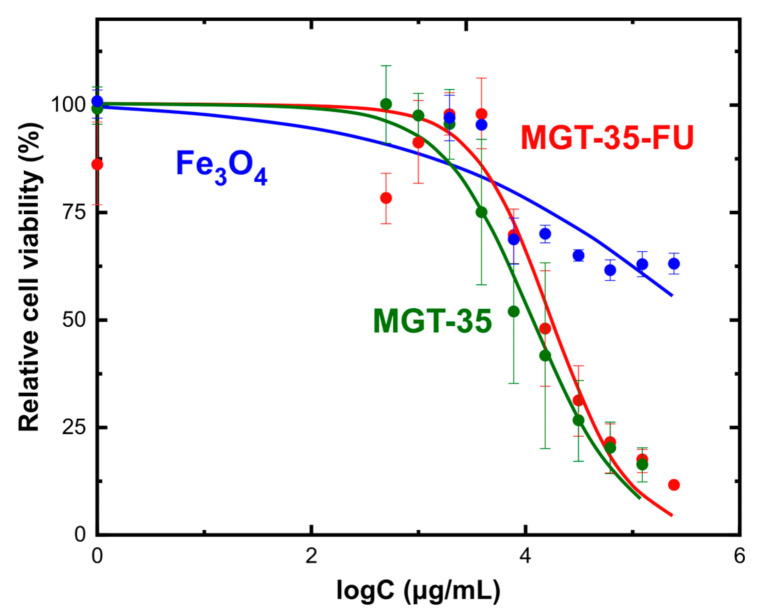
Cytotoxicity of HT29 cells exposed to Fe_3_O_4_ nanoparticles (blue line), MGT-35 (green line) and MGT-35-FU (red line). The cell viability was normalized using the highest signal measured and the percentage was plotted against the logarithm of the samples’ concentration.

## Data Availability

Please refer to suggested Data Availability Statements in section “MDPI Research Data Policies” at https://www.mdpi.com/ethics (accessed on 8 July 2021).

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
