# Peer review of "Hydrotalcite-Embedded Magnetite Nanoparticles for Hyperthermia-Triggered Chemotherapy"

_nanomaterials, 2021, doi:10.3390/nano11071796_

Round 1
Reviewer 1 Report
The manuscript entitled “Hydrotalcite-embedded magnetite nanoparticles for hyperthermia-triggered chemotherapy” investigated the magnetic nanocomposite, consisting of Fe3O4 nanoparticles and 5-fluorouracil molecules embedded into a Mg/Al layered double hydroxide matrix. The authors provide easy and scalable procedure with continuous-flow reactors using for production of the drug-delivery system. The topic is relevant to the journal Nanomaterials, and the content presents interesting results. However, several issues need to be addressed before publication.
- Line 56: What is ' buffering activity of the medium'? Please explain.
- Lines 263-264: The authors' claim about “metallic Fe core” is questionable. The authors need to provide some supporting evidence if available.
- Lines 317-320: authors should submit the formula of 5-fluorouracil and the reaction between the nanocomposite and 5-fluorouracil. Besides, the loss of carbonate content at the end of this experiment is only 1 % wt. It appears to be very small value. What is the experimental error?
- Lines 328-329: the release of the drug triggered by modifying the pH of the dispersion medium. Please explain the process.
- Lines 336-340: It should be explained why such temperature range (10-35 °C) was chosen for the study. This temperature is much lower than body temperature. Please explain the effect of temperature on the 5-fluorouracil
- 9: What are the reasons for the IC50 values of MGT-35 (12 mg/mL) is lower than MGT-35-FU (18 mg/mL)? And is there any reason to use 5-fluorouracil? What about IC50 values of Mg/Al layered double hydroxide matrix?
- As for me, an improvement of the discussion is needed in relation to similar or analogous results from literature.
Author Response
The manuscript entitled “Hydrotalcite-embedded magnetite nanoparticles for hyperthermia-triggered chemotherapy” investigated the magnetic nanocomposite, consisting of Fe3O4 nanoparticles and 5-fluorouracil molecules embedded into a Mg/Al layered double hydroxide matrix. The authors provide easy and scalable procedure with continuous-flow reactors using for production of the drug-delivery system. The topic is relevant to the journal Nanomaterials, and the content presents interesting results. However, several issues need to be addressed before publication.
We would like to thank the reviewer for the good impression on the manuscript and the understanding on the main advantages of the developed approach.
Line 56: What is ' buffering activity of the medium'? Please explain.
Indeed, this was not clear. We refer to the pH buffering property of biological solutions that in combination to the body temperature facilitate the continuous dissolution of the components in nanoparticles in relatively short periods.
Changes in manuscript
Line 55
and degradation effects triggered by the dissolution activity of buffer solutions on solid phases.
Lines 263-264: The authors' claim about “metallic Fe core” is questionable. The authors need to provide some supporting evidence if available.
We must apologize for this mistake. We are sorry that the whole paragraph referring to this point does not belong to this work but accidentally remained during transfer to the journal’s template. We hope that both reviewers and editors will understand that this paragraph has nothing to do with the current manuscript.
Changes in manuscript
Line 260
XRD diagrams of the prepared samples are shown in Figure 1 for comparison. As illustrated, both systems are composed of metal iron and an iron oxide with inverse spinel structure e.g., Fe3O4. The intensity of identified peaks corresponding to each phase comes in accordance to the core-shell configuration derived through TEM observations (Figure 2). In particular, spherical NPs have an average diameter of around 35 nm with relatively high polydispersity (~20 %). A low-contrast layer that surrounds most of the particles (around 2.5 nm in thickness) provides evidence for the assumed iron core-iron oxide shell morphology. This oxide shell engulfing the metallic Fe core is polycrystalline.
Lines 317-320: authors should submit the formula of 5-fluorouracil and the reaction between the nanocomposite and 5-fluorouracil. Besides, the loss of carbonate content at the end of this experiment is only 1 % wt. It appears to be very small value. What is the experimental error?
According to the suggestion from the reviewer we included the formula and the exchange reaction with the layered double hydroxide. Concerning the percentage of carbonate losses in the exchange procedure with 5-fluorouracil, we believe that 1 %wt. is not a small but, on the opposite, a large percentage with respect to the occurring concentration of the drug in the solution. In particular, considering the achieved uptake capacity of the drug, these losses correspond to more than 50 times higher values than the expected according to the stoichiometry of the ion exchange. This suggests that the incorporation of drug’s voluminous molecule causes the release of multiple carbonate ions in order to fit in the interlayer space. The experimental error of carbonate detection method is less than 5 %.
Changes in manuscript
Line 104
loaded with anticancer 5-fluorouracil (C4H3FN2O2, FU)
Line 328
The procedure is described as:
Mg6Al2(OH)16(CO32-) + C4H3FN2O2- → Mg6Al2(OH)16(C4H3FN2O2-) + CO32-
Line 332
Considering the uptake capacity values for 5-fluorouracil, carbonate losses appear much higher than the stoichiometrically expected ones, suggesting that the incorporation of drug’s voluminous molecule causes the release of multiple carbonate ions in order to fit in the interlayer space.
Line 165
The percentage of carbonates (CO32-) located in the interlayer space of hydrotalcite was quantified using a FOGL bench-top soil calcimeter with a determination error of less than 5 %.
Lines 328-329: the release of the drug triggered by modifying the pH of the dispersion medium. Please explain the process.
In this case, the stability of 5-fluouracil in the structure mainly depends on the solubility of the layered double hydroxide components which generally increases as we move to pH values below 9. We assume that the weakening of the layered structure by the departure of Mg and Al ions is the reason for observed drug release.
Changes in manuscript
Line 345
The weakening of the layered structure by the increase of metal components solubility is the reason for increasingly observed drug release.
Lines 336-340: It should be explained why such temperature range (10-35 °C) was chosen for the study. This temperature is much lower than body temperature. Please explain the effect of temperature on the 5-fluorouracil
The main aim of the temperature dependence study was to get a general view of the release kinetics in a wide temperature range rather than simulating the body temperature conditions. Practically, the 35 oC case is very close to the body temperature and the release curve may provide an impression of the expected behavior.
The findings indicate that the release of drug is proportional to the temperature increase. Again, the increase of solubility of the layered double hydroxide components is the key parameter. However, it is important that even at temperatures close to the human body one, around 20 % can remain loaded on the nanocomposite for many hours. The data can be a guide on what we should expect during the storage of the dispersion in a refrigerator and the behavior during its application for cancer treatment. All these in comparison to the sudden release observed under magnetic hyperthermia conditions which is the main finding of the work.
Changes in manuscript
Line 357
More specifically, the loading loss at temperature 10, 20 and 35 oC (close to the human body temperature) after 5 h of contact at pH 7.4 was found to be 45, 55 and 70 %wt. respectively. The temperature dependence study provides a general view of the release kinetics in a wide temperature range indicating the behavior of the drug-loaded nanocomposite during storage in a refrigerator or its application in cancer treatment. Importantly, even at the higher temperature, around 20 % of the initial load is stabilized into the nanocomposite for many hours.
9: What are the reasons for the IC50 values of MGT-35 (12 mg/mL) is lower than MGT-35-FU (18 mg/mL)? And is there any reason to use 5-fluorouracil? What about IC50 values of Mg/Al layered double hydroxide matrix?
We must specify that the MGT-35 sample corresponds to the pure Mg/Al layered double hydroxide. Indeed, it is a tricky result that loaded and pure nanocomposites provide the same toxicity in cells. We would say that during the test, the toxicity of the Mg/Al layered double hydroxide matrix dominates in both samples and the presence of the 5-fluorouracil is minor.
We believe that during previous washing of the nanocomposite the expected drug release of around 80 % takes place before contact with cells and most of the remaining 20 % is stabilized into the structure for the time window of the experiment, as explained by the temperature dependent kinetic data. The application of localized heating by magnetic hyperthermia is in position to overcome this barrier and rapidly trigger the complete release of this fraction which appears to be the most strongly captured.
As already mentioned in the manuscript, the fact that a protein corona is formed upon dispersing the nanocomposite in cell growth medium is expected to influence the drug release and may introduce some kind of interference in the short-term toxicity kinetics. This is something we examine this period in a wide variety of nanoparticles used for biomedical applications.
Changes in manuscript
Line 410
Noteworthy, MGT-35-FU and MGT-35 exhibited similar IC50 values. Their toxic profile appears to be dominated by the Mg/Al layered double hydroxide matrix while the effect of the 5-fluorouracil presence is minor. Such finding is attributed to the drug release of around 80 % (Figure 7) which takes place during preliminary washing of the nanocomposite before contacting with cells. The remaining 20 % can be stabilized into the structure for the time window of the experiment, as explained by the temperature dependent kinetic data. The application of localized heating by magnetic hyperthermia is in position to overcome this barrier and rapidly trigger the complete release of this fraction which appears to be the most strongly captured. Another possible explanation is that the presence of serum protein corona around the nanoparticles may influence the short-term release kinetics of 5-fluorouracil.[35,38]
As for me, an improvement of the discussion is needed in relation to similar or analogous results from literature.
This is correct. We tried to cover this part by adding relevant literature results for comparison.
Changes in manuscript
Line 307
Compared to other studies on single iron oxide nanoparticles or LDH composites, the obtained SAR values are among the highest reported covering a very wide frequency range. Suggestively, Fe3O4 nanoparticles prepared by the oxidative precipitation method, showed efficiency of around 2 kW/g at 765 kHz, translated into 2.3 W/gFe when produced by a continuous flow process [34], but less than the half when produced in batch [35]. In order to achieve higher SAR values (up to 10 kW/g at 500 kHz), combined ferrite phases were employed but the requirement for using hazardous reagents for their synthesis and the toxicity of elements such as Mn and Co inhibit their potential for clinical application [36–38]. Research about the heating performance of magnetic nanoparticles-decorated LDH systems is limited. In an example, Fe3O4/Mg-Al LDH nanohybrids were found to reach a SAR of 73.5 W/g at 425 kHz and 30 kA/m [31]. The same study provides promising results concerning the combined hyperthermia and drug delivery with doxorubicin as well as the therapeutic efficiency in HeLa cells.

Reviewer 2 Report
The manuscript is correctly written in general, and it is easy to read. The proposed particles appear to be interesting for drug delivery in combination with hyperthermia, although it is not clear to what extent they are advantageous over simpler magnetic nanostructures. Thi is an aspect of the manuscript that should be improved, as I mention below:
- The synthesis and structure of the proposed nanoparticles is more complex that previously reported ones. The authors should perhaps stress their advantages over existing nanostructures.
- Can the synthesis be simplified by running it in successive batches instead of using the continuous flow method? Or is it rather essential to use the latter?
- Line 177 page 4: the authors forget to include the model and manufacturer of the hyperhermis device
- Lines 257 and 260: Figure numbers are erroneous (1 and 2 are in reality 3 and 4)
- The obtained particles appear in Fig. 4 to be very large aggregates. Can't we have a less magnification picture in order to appreciate the actual structure of the synthesized particles?
- Line 261: instead of providing approximate polydispersity, it would be preferable to have mean +- S.D. of all samples
- The SAR obtained is certainly high. It might be compared with reported values. Also, a +- uncertainty should be provided. Some example of heating plots should be provided. Finally, it is stated that the temperature reaches 35 ºC, but it refers obviously to temperature INCREASE.
- I understand that 5-FU is a drug, difficult to dissolve in water. Did you find any problems with this?
Author Response
The manuscript is correctly written in general, and it is easy to read. The proposed particles appear to be interesting for drug delivery in combination with hyperthermia, although it is not clear to what extent they are advantageous over simpler magnetic nanostructures. This is an aspect of the manuscript that should be improved, as I mention below:
We thank the reviewer for finding the manuscript interesting.
The synthesis and structure of the proposed nanoparticles is more complex that previously reported ones. The authors should perhaps stress their advantages over existing nanostructures.
As explained in the manuscript, the main advantage of the proposed nanocomposite is its fully inorganic nature which is able to combine the heating capability of magnetic nanoparticles with the drug hosting capacity of the layered double hydroxide as well as their potential collaboration in combined thermal/chemical treatment to cancer cells. Considering that the Mg/Al layered double hydroxide is already recognized for its compatibility for human use as an antacid, this approach brings an alternative to the typical nanoparticles-based drug carriers which are based on the use of surface-attached ligands most of them related to some kind of toxicity.
As explained in the next question, the synthesis of the nanocomposites is not a complicated one since it stands on metal salts aqueous precipitation, one of the easiest chemical methods to obtain nanoparticles. The fact that the method is adopted to a continuous flow system is just a way to succeed reproducibility and constant conditions while it indicates how to minimize the operation cost and scale-up to mass production.
Changes in manuscript
Line 108
The main advantage of the proposed nanocomposites is their fully inorganic nature which is able to combine the heating capability of magnetic nanoparticles with the drug hosting capacity of the layered double hydroxide considering also that the Mg/Al layered double hydroxide is already recognized for its compatibility for human use as an antacid.
Can the synthesis be simplified by running it in successive batches instead of using the continuous flow method? Or is it rather essential to use the latter?
The answer in the first question is yes. Obviously, the sequential salts precipitation reactions can take place in batch mode and provide a final product similar to the described in this work. However, we would like to promote the significant advantages of the continuous flow variant which ensures the achievement of constant concentrations in all ionic and solid forms throughout the production line when steady-state is reached, the possibility to control critical parameters through on-line regulation of synthesis parameters such as the reactor’s pH, and enable high production capacities, low energy consumption and proportional scale-up at any volume.
Changes in manuscript
Line 132
The described reactions can be realized with similar success in batch reactors, however, advantages such as the good reproducibility, the achievement of constant concentrations in all ionic and solid forms, the minimization of operation cost and the scale-up potential to mass production would not be covered.
Line 177 page 4: the authors forget to include the model and manufacturer of the hyperthermia device
Yes, this information was included in the revised version.
Changes in manuscript
Line 176
Calorimetry measurements of magnetic suspensions under HAC were performed using a commercial AC magnetic field generator (SPG–06-III 6 kW High Frequency Induction Heating Machine, Shenzhen Shuangping Ltd.) working at 765 kHz frequency and 24 kA/m magnetic field intensity.
Lines 257 and 260: Figure numbers are erroneous (1 and 2 are in reality 3 and 4)
This inconsistency is the result of our mistake to include a whole paragraph not belonging to this work but accidentally remained during transfer to the journal’s template.
Once more, we must apologize for this mistake.
Changes in manuscript
Line 260
XRD diagrams of the prepared samples are shown in Figure 1 for comparison. As illustrated, both systems are composed of metal iron and an iron oxide with inverse spinel structure e.g., Fe3O4. The intensity of identified peaks corresponding to each phase comes in accordance to the core-shell configuration derived through TEM observations (Figure 2). In particular, spherical NPs have an average diameter of around 35 nm with relatively high polydispersity (~20 %). A low-contrast layer that surrounds most of the particles (around 2.5 nm in thickness) provides evidence for the assumed iron core-iron oxide shell morphology. This oxide shell engulfing the metallic Fe core is polycrystalline.
The obtained particles appear in Fig. 4 to be very large aggregates. Can't we have a less magnification picture in order to appreciate the actual structure of the synthesized particles?
In the TEM study, our intention was to include some representative views of the nanocomposite at the thinner sides of the main volume where electron beam transmission was possible and mainly show the distribution of magnetite nanoparticles which indeed, show aggregation since no stabilizing agent was added during their synthesis (if this is what the reviewer refers to). In order to avoid misunderstandings, the studied nanocomposite consists of magnetite nanoparticles distributed in a layered double hydroxide which is organized in a continuous network with relatively large dimensions. Suggestively, the average hydrodynamic diameter of the nanocomposite was found to be around 500 nm. However, such impression could not be reflected in the electron microscopy images which correspond to the dried condition of the nanocomposites. Lower magnification images of the sample MGT-35 taken with scanning electron microscopy, were added in the Supporting information file indicating that in the powder form, the layered matrix appears as a continuous substrate without obvious limits of the separation that occurs when dispersed. We believe that this effect is attributed to the secondary self-organization of the layered structure by weak forces during dewatering and it is fully reversible.
Changes in manuscript
Line 279
Lower magnification images of the sample MGT-35 (Figure S7), taken with scanning electron microscopy, indicate that in the powder form, the layered matrix appears as a continuous substrate without obvious limits of the separation that occurs when dispersed. This effect may be attributed to the secondary self-organization of the layered structure by weak forces during dewatering and it is fully reversible considering that a hydrodynamic diameter of samples was measured around 500 nm.
Changes in Supporting information
Figure S7. Scanning electron microscopy images and elemental analysis of sample MGT-35.
Line 261: instead of providing approximate polydispersity, it would be preferable to have mean +- S.D. of all samples
This point also refers to the presence of a paragraph without meaning to this manuscript.
Changes in manuscript
Line 260
XRD diagrams of the prepared samples are shown in Figure 1 for comparison. As illustrated, both systems are composed of metal iron and an iron oxide with inverse spinel structure e.g., Fe3O4. The intensity of identified peaks corresponding to each phase comes in accordance to the core-shell configuration derived through TEM observations (Figure 2). In particular, spherical NPs have an average diameter of around 35 nm with relatively high polydispersity (~20 %). A low-contrast layer that surrounds most of the particles (around 2.5 nm in thickness) provides evidence for the assumed iron core-iron oxide shell morphology. This oxide shell engulfing the metallic Fe core is polycrystalline.
The SAR obtained is certainly high. It might be compared with reported values. Also, a +- uncertainty should be provided. Some example of heating plots should be provided. Finally, it is stated that the temperature reaches 35 ºC, but it refers obviously to temperature INCREASE.
A discussion on the comparison of SAR values with literature values was introduced in the revised version. Error bars on the SAR estimations are included in the points of Figure 6. At the lower frequencies range, the SAR error is quite small (typically below 3 %) following the accuracy of the AC magnetometry measurements while for the frequency of 765 kHz where an induction heater was used, error bar is around ±10 %.
Examples of heating plots, i.e. temperature versus time curves, which were used to estimate SAR values, were already included in the Supporting information. We also tried to rephrase the sentence clarifying that in this example the temperature rises from room temperature to 35 oC.
Changes in manuscript
Line 297
For instance, a temperature increase from room temperature to around 35 oC within 2 min of field application was obtained for a 2 g/L aqueous dispersion of sample MGT-35 corresponding to a SAR value of 1970 W/gFe.
Line 301
The uncertainty of determined SAR values in the lower frequencies range is quite small (typically below 3 %) following the accuracy of the AC magnetometry measurements.
Line 307
Compared to other studies on single iron oxide nanoparticles or LDH composites, the obtained SAR values are among the highest reported covering a very wide frequency range. Suggestively, Fe3O4 nanoparticles prepared by the oxidative precipitation method, showed efficiency of around 2 kW/g at 765 kHz, translated into 2.3 W/gFe when produced by a continuous flow process [34], but less than the half when produced in batch [35]. In order to achieve higher SAR values (up to 10 kW/g at 500 kHz), combined ferrite phases were employed but the requirement for using hazardous reagents for their synthesis and the toxicity of elements such as Mn and Co inhibit their potential for clinical application [36–38]. Research about the heating performance of magnetic nanoparticles-decorated LDH systems is limited. In an example, Fe3O4/Mg-Al LDH nanohybrids were found to reach a SAR of 73.5 W/g at 425 kHz and 30 kA/m [31]. The same study provides promising results concerning the combined hyperthermia and drug delivery with doxorubicin as well as the therapeutic efficiency in HeLa cells.
I understand that 5-FU is a drug, difficult to dissolve in water. Did you find any problems with this?
Indeed, the solubility of 5-fluorouracil in water is relatively low. In our case, the nominal solubility in PBS is 8 mg/mL or around 60 mM. For this reason, the used stock solution which was regularly prepared in the same day with the experiments, had a concentration of 20 mM.
Changes in manuscript
Line 203
In these experiments, a freshly prepared 20 mM stock solution of 5-fluorouracil in PBS was used after proper dilution.
